# Generator Born from Classifier

**Runpeng Yu**  **Xinchao Wang**[†]
National University of Singapore
r.yu@u.nus.edu   xinchao@nus.edu.sg

## Abstract

In this paper, we make a bold attempt toward an ambitious task: given a pre-trained classifier, we aim to reconstruct an image generator, without relying on any data samples. From a black-box perspective, this challenge seems intractable, since it inevitably involves identifying the inverse function for a classifier, which is, by nature, an information extraction process. As such, we resort to leveraging the knowledge encapsulated within the parameters of the neural network. Grounded on the theory of Maximum-Margin Bias of gradient descent, we propose a novel learning paradigm, in which the generator is trained to ensure that the convergence conditions of the network parameters are satisfied over the generated distribution of the samples. Empirical validation from various image generation tasks substantiates the efficacy of our strategy.

## 1 Introduction

The majority of machine learning research bifurcates into two distinct branches of study: the predictive task and the generative task. Given the input $\boldsymbol{x}$ and the label $y$, the former one focuses on the training of a high-performing classifier or regressor, which approximates $p(y|\boldsymbol{x})$ [Vaswani et al., 2023, Dosovitskiy et al., 2021, Jing et al., 2023], whereas the latter one aims to train a generative model capable of sampling from $p(\boldsymbol{x}|y)$ or $p(\boldsymbol{x}, y)$ [Goodfellow et al., 2014]. The gap between the predictive and the generative models, as a result, predominantly arises from the lack of information in the predictive models about the marginal distribution $p(\boldsymbol{x})$. In the realm of deep neural networks, however, the over-parameterization leads to the overfitting on the training distribution and the memorization of the training samples [Feldman and Zhang, 2020, Daniely, 2020, Arpit et al., 2017], which, in turn, make the network implicitly retain information about $p(\boldsymbol{x})$. With this component in hand, it prompts the question of whether it is feasible to derive a generative model from a predictive one.

In this paper, we explore this novel task, which attempts to learn a generator directly from a pre-trained classifier, *without* the assistance of any training data. Unarguably, this is a highly ambitious task with substantial difficulty, as either explicitly extracting information about $p(\boldsymbol{x})$ from a pre-trained classifier or directly solving this inverse problem from classifier to generator poses significant challenges. Despite these challenges, the value of this task lies in its potential to offer a new approach to training generators that mitigates the direct dependence on large volumes of training data. This provides a possible solution for learning tasks in scenarios where data is scarce or unavailable. Moreover, this task presents a novel way to utilize and analyze the pre-trained predictive models, facilitating our understanding of the encoded information within the parameters.

To this end, we propose a novel learning scheme. Our approach is grounded in the theory of Maximum-Margin Bias of gradient descent, which demonstrates that the parameters of a neural network trained via gradient descent will converge to the solution of a specific optimization problem. This optimization problem minimizes the norm of the neural network parameters while maximizing the classification margin on the training dataset. The necessary condition for the solution of this

---

[†] Corresponding author.

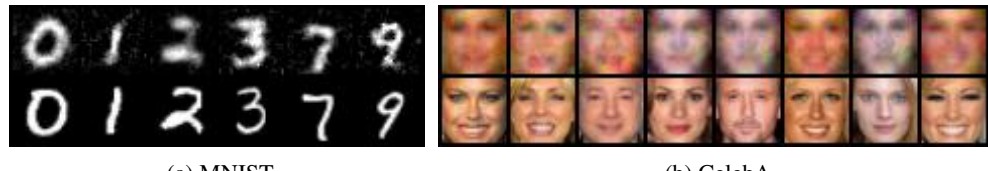

(a) MNIST            (b) CelebA

Figure 1: Images produced by the generator trained only using pre-trained classifier. The generated images, positioned in the first row, are accompanied by their nearest neighbors from the original dataset, displayed in the second row.

optimization problem constructs a system of equations, describing the relationship between the pre-trained neural network parameters and the training data distribution.

Since our aim is to learn a generator from the pre-trained classifier parameters, the generator is therefore expected to approximate a distribution that satisfies the necessary condition of this optimization problem. To accomplish this, we design the loss function for training the generator and the corresponding training algorithm based on the necessary condition. The entire training process does not rely on any training data; all available information related to the pre-trained data is encapsulated within the parameters of the pre-trained classifier.

The intuition behind our design is twofold: on one hand, the generator should guarantee that the pre-trained classifier performs well under the data distribution it approximated, and on the other hand, the generator should ensure that the current classifier parameters are the convergence point of the gradient descent algorithm under the data distribution it approximated. It's noteworthy that the original data distribution naturally satisfies these conditions. Therefore, we anticipate that employing the proposed method will guide the generator to discover the original data distribution.

We conduct experiments on commonly used image datasets. Fig. 1 shows some generated images of the MNIST and CelebA datasets. Remarkably, even trained without access to the original data, the generator is able to perform conditional sampling and generate the digits and faces.

Our contribution is therefore a novel approach that, for the first time, attempts to train a generator from a pre-trained classifier without utilizing training data. The proposed approach produces encouraging results on synthetic and real-world images.

A list of symbols utilized in this paper and corresponding descriptions can be found in the Appendix.

## 2 Related Work

**Generative Adversarial Networks.** Generative Adversarial Network (GAN) consists of a generator and a discriminator collectively optimizing a minimax problem to learn and replicate the original data distribution [Goodfellow et al., 2014]. Numerous extensions of the original GAN have been investigated, including functionality enhancement [Chen et al., 2016, Mirza and Osindero, 2014, Odena et al., 2017, Donahue et al., 2017]; architecture optimization and scaling [Karras et al., 2018, Brock et al., 2019, Denton et al., 2015]; and training loss design [Nowozin et al., 2016, Arjovsky et al., 2017, Wei et al., 2018]. Owing to their superior generation quality, GANs have found wide-ranging applications in image synthesis [Yang et al., 2017b], blending [Wu et al., 2017], inpainting [Yeh et al., 2017, Yang et al., 2017a, Yu et al., 2018], super-resolution [Ledig et al., 2017], denoising [Linh et al., 2020]; image-to-image translation [Zhu et al., 2017, Isola et al., 2017]; 3D object generation [Wu et al., 2016]; video generation [Vondrick et al., 2016, Tulyakov et al., 2018]; etc.

Both our work and GAN require an additional classifier to guide the training of the generation and provide a measure of authenticity for generated data. However, in the GAN framework, the classifier is trained concurrently with the generator, with the explicit goal of discerning the quality of generated results. In contrast, our method utilizes a pre-trained classifier, which can be arbitrary, and its training objective is to maximize classification accuracy, not to judge the quality of generated results. This imbues our approach with considerable flexibility. Furthermore, during the training of GAN, the generator has access to training data. However, in our task, the training data is not available to the generator, which makes our task harder.

**Feature Visualization and Model Inversion.** Besides our task, neural network feature visualization and model inversion share the objective of extracting information relevant to the training data from

pre-trained classifiers. Neural network feature visualization is a technique aimed at identifying input data that maximally activates specific neurons or layers within the network, thereby providing insights into the patterns or features that the network is primed to recognize. [Engstrom et al., 2019, Olah et al., 2017, Nguyen et al., 2016] Model inversion in neural networks refers to the process of inferring or reconstructing input data given the trained model. [Yin et al., 2020] In the realm of adversarial attacks, model inversion is deployed to discover sensitive information about the training data from the model's outputs. [He et al., 2019, Zhao et al., 2021, Fredrikson et al., 2015]

Unlike these tasks, where an independent gradient optimization is required for each generation, our goal is to develop a generator capable of sampling from the training data distribution. While there is research in neural network feature visualization and model inversion that utilizes a generative model to assist in the restoration of specific training data, these works typically employ the generator more as a prior for the recovery process. [Nguyen et al., 2017, Jeon et al., 2021, Yang et al., 2019, Zhang et al., 2020] The training of such a generator necessitates additional training data, which should be similar to or encompass the original training data used for the classifier. In contrast, our approach directly derives a generator from the classifier without the utilization of extraneous training data.

**Energy-Based Model.** Within the framework of energy-based models, pre-trained classifiers have also been demonstrated to be capable of acting as a conditional probability distribution, generating data samples. [LeCun et al., 2006, Grathwohl et al., 2020, Guo et al., 2023] Although the energy-based models theoretically bridge predictive and generative models, practical implementation for sample generation based on it still relies on an optimization target and executes a multi-step gradient optimization process, akin to model inversion. In contrast, our objective is to develop a generator with the ability for random or condition-based sampling, procuring data samples directly through the forward pass of the neural network.

**Maximum-Margin Bias of Gradient Descent.** Our work is based on the study of the Maximum-Margin Bias of gradient descent. These investigations primarily seek to elucidate why gradient descent algorithms are capable of learning models with robust generalization capabilities, even in the absence of additional regularization constraints. It has been discovered that under the guidance of gradient descent, the parameters of a neural network converge to a solution to an optimization problem aiming to maximize the classification boundary while concurrently minimizing the norm of the network parameters. Initial studies focused on linear logistic regression models [Rosset et al., 2003, Soudry et al., 2018], which were subsequently extended to homogeneous neural networks [Wei et al., 2019, Xu et al., 2018, Nacson et al., 2019, Lyu and Li, 2020b, Ji and Telgarsky, 2020, Le and Jegelka, 2022] and, more recently, to a broader class of quasi-homogeneous neural networks [Kunin et al., 2023]. The theory of Maximum-Margin Bias has also been leveraged by Haim et al. [2022] to recover training data. However, their objective was the restoration of data rather than training a generator, and their work was exclusively confined to binary classification datasets and fully connected networks without bias terms.

# 3 Preliminary

To extract the information about the training dataset from the parameters of the pre-trained classification model, we leverage the theory of Maximum-Margin Bias.

Let $\Phi(\cdot; \zeta) : \mathcal{R}^d \to \mathcal{Y}$ denote the classifier parameterized by $\zeta$ and trained on multi-class classification dataset $D = \{(\boldsymbol{x}_i, y_i)\}_{i=1}^N$ with each $(\boldsymbol{x}_i, y_i) \in \mathcal{R}^d \times \mathcal{Y}$. let $L(\zeta) := \sum_{i=1}^N l(\Phi(\boldsymbol{x}_i; \zeta), y_i)$ denote the standard cross-entropy loss of $\Phi$ on $D$. To extend the applicability of Maximum-Margin Bias analysis to various neural network structures, the definition of the $\Lambda$-*quasi-homogeneous model* is introduced. For a (non-zero) positive semi-definite diagonal matrix $\Lambda$, a model $\Phi(\cdot; \zeta)$ is $\Lambda$-quasi-homogeneous if the model output scales as $\Phi(\boldsymbol{x}; \psi_\alpha(\zeta)) = e^\alpha \Phi(\boldsymbol{x}; \zeta)$ for all $\alpha \in \mathcal{R}$ and input $\boldsymbol{x}$, when the parameter scales as $\psi_\alpha(\zeta) := e^{\alpha \Lambda} \zeta$. Many commonly used neural network architectures satisfy the definition of $\Lambda$-quasi-homogeneous model, such as convolutional neural networks and fully connected networks with biases, residual connections, and normalization layers. The seminorm of $\zeta$ corresponding to $\Lambda$ is defined as $||\zeta||_\Lambda^2 := \zeta^T \Lambda \zeta$. Let $\lambda_{max} := \max_j \Lambda_{jj}$ denote the maximal element in $\Lambda$, $\tilde{\Lambda}$ is the matrix setting all elements less than $\lambda_{max}$ to 0, *i.e.*, $\tilde{\Lambda}_{jj} = \lambda_{max}$ if $\Lambda_{jj} = \lambda_{max}$, otherwise, $\tilde{\Lambda}_{jj} = 0$. Accordingly, the seminorm of $\zeta$ corresponding to $\tilde{\Lambda}$ is defined as $||\zeta||_{\tilde{\Lambda}}^2 := \zeta^T \tilde{\Lambda} \zeta$. The

normalized parameters is defined as $\bar{\zeta} := \psi_\tau(\zeta)$, such that $\|\bar{\zeta}\|_\Lambda^2 = 1$. The Quasi-Homogeneous Maximum-Margin Theorem states as follows.

**Theorem 1** (Paraphrased from [Kunin et al., 2023]). *Let $\Phi(\cdot; \zeta)$ denote a $\Lambda$-quasi-homogeneous classifier trained on $D$ with cross-entropy loss $L$. Assume that: (1) for any fixed $\boldsymbol{x}$, $\Phi(\boldsymbol{x}; \zeta)$ is locally Lipschitz and admits a chain rule [Davis et al., 2020, Lyu and Li, 2020a]; (2) the learning dynamic is described by a gradient flow [Lyu and Li, 2020a]; (3) $\lim_{t\to\infty} \bar{\zeta}(t)$ exists; (4) $\exists \kappa > 0$ such that only $\zeta$ with $\|\zeta\|_{\Lambda_{\max}} \geqslant \kappa$ separates the training data; and (5) $\exists t_0$ such that $L(\zeta(t_0)) < N^{-1} \log 2$. $\exists \alpha \in \mathcal{R}$ such that $\tilde{\zeta} := \psi_\alpha(\lim_{t\to\infty} \bar{\zeta}(t))$ is a first-order stationary point of the following maximum-margin problem*

$$\min_{\zeta'} \quad \frac{1}{2}\|\zeta'\|_\Lambda^2 \tag{1a}$$

$$s.\,t. \quad \min_{c \in \mathcal{Y}/\{y_i\}} \Phi_{y_i}(x_i; \zeta') - \Phi_c(x_i; \zeta') \geqslant 1 \quad \forall i \in [N], \tag{1b}$$

*where $\Phi_c(\cdot; \zeta')$ is the prediction of $\Phi$ for the class $c \in \mathcal{Y}$.*

## 4  Method

Theorem 1 implies that the neural network parameters converge to the first-order stationary point (or the Karush–Kuhn–Tucker point (KKT) point) of the optimization problem in Eq. (1). Let $\{\mu_{ic}\}_{i\in[N],c\in\mathcal{Y}/\{y_i\}}$ denote the set of KKT multipliers, the KKT condition can be written as follows.

$$\tilde{\Lambda}\tilde{\zeta} = \sum_{i\in[N]} \sum_{c\in\mathcal{Y}/\{y_i\}} \mu_{ic}[\nabla_{\tilde{\zeta}}\Phi_c(x_i; \tilde{\zeta}) - \nabla_{\tilde{\zeta}}\Phi_{y_i}(x_i; \tilde{\zeta})]; \tag{2a}$$

for all $i \in [N]$, and $c \in \mathcal{Y}/\{y_i\}$ :

$$\Phi_{y_i}(x_i; \tilde{\zeta}) - \Phi_c(x_i; \tilde{\zeta}) \geqslant 1, \tag{2b}$$

$$\mu_{ic} \geqslant 0, \tag{2c}$$

$$\mu_{ic}[1 + \Phi_c(x_i; \tilde{\zeta}) - \Phi_{y_i}(x_i; \tilde{\zeta})] = 0, \tag{2d}$$

where the Eqs. (2a) to (2d) are known as the stationarity condition, primal feasibility condition, dual feasibility condition, and the complementary slackness condition, respectively.

The intuition behind the stationarity condition can be summarized as follows. According to the value of the corresponding element in $\Lambda$, the neural network parameters can be divided into two groups: the first group $Z_1 := \{\tilde{\zeta}_j | \Lambda_{jj} = \lambda_{max}\}$ includes all the parameters whose corresponding elements in $\Lambda$ are equal to $\lambda_{max}$, and the second set $Z_2 := \{\tilde{\zeta}_j | \Lambda_{jj} \neq \lambda_{max}\}$ contains all the parameters whose corresponding elements in $\Lambda$ are not equal to $\lambda_{max}$. The stationarity condition in Eq. (2a) evaluates the linear combination of the derivatives of the neural network output corresponding to the parameters, with KKT multipliers $\{\mu_{ic}\}_{i\in[N],c\in\mathcal{Y}/\{y_i\}}$ as coefficients of the combination. Such a linear combination of the derivatives corresponding to the parameters in $Z_1$ is equal to the parameters in $Z_1$. In contrast, such a linear combination of the derivatives corresponding to the parameters in $Z_2$ is equal to a zero vector.

The intuition behind other conditions can be summarized as follows. According to the complementary slackness condition in Eq. (2d), $\mu_{ic}$ is nonzero only when $\Phi_{y_i}(x_i; \tilde{\zeta}) - \Phi_c(x_i; \tilde{\zeta}) = 1$. Two conditions are required for $\mu_{ic}$ to be nonzero. First, for a pair of $(i, c)$, $\mu_{ic}$ can only be nonzero when $c$ is the class with the second largest predicted probability for the sample $x_i$, *i.e.*, $c \in \mathcal{S}_i$, where $\mathcal{S}_i := \{y|\Phi_y(x_i; \tilde{\zeta}) = \max_{y'\in\mathcal{Y}/\{y_i\}} \Phi_{y'}(x_i; \tilde{\zeta})\}$. Second, according to primal feasibility in Eq. (2b), the minimum possible value of $\Phi_{y_i}(x_i; \tilde{\zeta}) - \Phi_c(x_i; \tilde{\zeta})$ is 1. Therefore, for a pair of $(i, c)$, $\mu_{ic}$ can only be nonzero when the margin between the true class $y_i$ and the class with the second largest predicted probability for the sample $x_i$ is minimum.

The evaluation of $\nabla_{\tilde{\zeta}}\Phi(\cdot; \tilde{\zeta})$ and $\Phi(\cdot; \tilde{\zeta})$ requires to first scale all parameters of a pre-trained network. For the convenience of practical implementation and following derivation, we transform $\tilde{\zeta}$ back to $\zeta$ using the definition of the quasi-homogeneous function. The KKT conditions in Eq. (2) are rewritten

as:

$$\bar{\Lambda}\zeta = \sum_{i \in [N]} \sum_{c \in \mathcal{Y}/\{y_i\}} \mu_{ic}[\nabla_\zeta \Phi_c(x_i; \zeta) - \nabla_\zeta \Phi_{y_i}(x_i; \zeta)]; \tag{3a}$$

for all $i \in [N]$, and $c \in \mathcal{Y}/\{y_i\}$ :

$$\Phi_{y_i}(x_i; \zeta) - \Phi_c(x_i; \zeta) \geqslant e^{-\alpha}, \tag{3b}$$

$$\mu_{ic} \geqslant 0, \tag{3c}$$

$$\mu_{ic}[e^{-\alpha} + \Phi_c(x_i; \zeta) - \Phi_{y_i}(x_i; \zeta)] = 0, \tag{3d}$$

where the new scaling parameter $\bar{\Lambda} := \tilde{\Lambda}e^{\alpha(2\Lambda - \mathbf{I})}$.

**From KKT condition to loss function.** Given only the pre-trained neural network $\Phi(\cdot; \zeta)$, the undetermined parts in the KKT conditions in Eq. (3) include the KKT multipliers, a set of $(x, y)$ pairs, and constant $\alpha$. Regarding the KKT multiplier a predictable objective, We use a neural network to learn it. Given an approximate distribution of the discrete random variable $y$, we sample $y$ directly. Our goal is to train a conditional generator $g$ parameterized by $\theta$ to generate input $x = g(\epsilon, y; \theta)$ given the corresponding label $y$ and random noise $\epsilon$. We also treat $\alpha$ as a learnable parameter, which will be discussed in detail later. In the following paragraphs, we first discuss how to design the loss function for learning these parameters and training the generator.

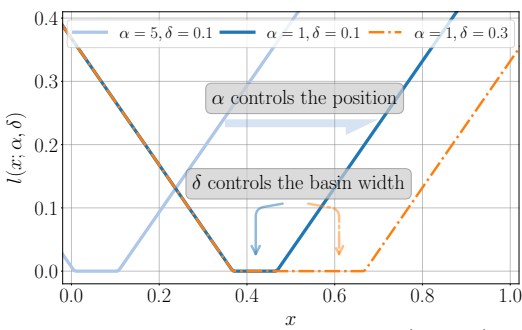

Figure 2: The U-shaped duality loss $l(x; \alpha, \delta) := \max(x - e^{-\alpha} - \delta, 0) - \min(x - e^{-\alpha}, 0)$.

First, we discuss how to ensure that the generated samples satisfy the stationary condition in Eq. (3a). The evaluation of the right-hand side of Eq. (3a) requires generating a dataset with a fixed number of samples every time to calculate the summation. Alternatively, we divide both sides of Eq. (3a) by the number of training samples $N$, which converts the sum on the right-hand side to an expectation:

$$\frac{1}{N}\bar{\Lambda}\zeta = \mathbb{E}_{x,y}\Big[\sum_{c \in \mathcal{Y}/\{y\}} \mu_c[\nabla_\zeta \Phi_c(x; \zeta) - \nabla_\zeta \Phi_y(x; \zeta)]\Big]. \tag{4}$$

Thus, according to the law of large numbers, the expectation can be estimated by the empirical average over a batch of $M$ random samples, where $M$ is a hyper-parameter. To ensure that the rescaled stationary condition in Eq. (4) can be satisfied, we use the following $L_{stationarity}$ to minimize the norm of the difference between both sides of Eq. (4).

$$L_{stationarity}(\theta, \eta) := ||\frac{1}{N}\bar{\Lambda}\zeta - \frac{1}{M}\sum_{i \in [M]} \sum_{c \in \mathcal{Y}/\{y_i\}} \mu_{ic}[\nabla_\zeta \Phi_{y_i}(x_i; \zeta) - \nabla_\zeta \Phi_c(x_i; \zeta)]||. \tag{5}$$

Conditions in Eqs. (3b) to (3d) constrain the valid values of the KKT multipliers. Accordingly, in order to satisfy the positivity condition of the KKT multipliers in dual feasibility, instead of directly optimizing the KKT multipliers, we use the proxy variables $\mu'_{ic}$, define $\mu_{ic} := ReLU(\mu'_{ic})$. For each generated sample $x_i$, we set up a $\mu'_i \in \mathcal{R}^{|\mathcal{Y}|}$ with $\mu'_{ic}$ is its $c$-th element. We learn $\mu'_i = h(x_i, y_i; \eta)$ by network $h$ parameterized by $\eta$. To approximate primal feasibility and complementary slackness, we require each generated sample $x_i$ to satisfy $0 \leqslant \Phi_{y_i}(x_i; \zeta) - \Phi_c(x_i; \zeta) - e^{-\alpha} \leqslant \delta$, where $0 < \delta \ll 1$ is a hyper-parameter to ensure numerical stability. We minimize $L_{duality}$ to approximate this constraint. Fig. 2 illustrates the shape of $L_{duality}$.

$$L_{duality}(\theta, \alpha) := \frac{1}{M}\sum_{i \in [M]} \sum_{c \in \mathcal{S}_i} [\max(\Phi_{y_i}(x_i; \zeta) - \Phi_c(x_i; \zeta) - e^{-\alpha} - \delta, 0)$$
$$- \min(\Phi_{y_i}(x_i; \zeta) - \Phi_c(x_i; \zeta) - e^{-\alpha}, 0)] \tag{6}$$

Final loss is the combination of $L_{lagrange}$ and $L_{duality}$ balanced by hyper-parameter $\beta$:

$$L = L_{stationarity}(\theta, \eta) + \beta L_{duality}(\theta, \alpha). \quad (7)$$

**Determine $\Lambda$.** To compute $L_{stationarity}$, it is necessary to determine the $\Lambda$ of the quasi-homogeneous function $\Phi(\cdot; \zeta)$. Here, we will introduce two properties of quasi-homogeneous functions and demonstrate how to construct a system of linear equations based on these properties to efficiently solve for $\Lambda$. First, taking the derivative of $\Phi(\boldsymbol{x}; \psi_\alpha(\zeta))$ corresponding to $\zeta$, we have

$$\nabla_{\psi_\alpha(\zeta)}\Phi(\boldsymbol{x}; \psi_\alpha(\zeta)) = e^{\alpha(\mathbf{I}-\Lambda)}\nabla_\zeta\Phi(\boldsymbol{x}; \zeta). \quad (8)$$

This indicates that, for any parameter $\zeta_i$ whose corresponding $\Lambda_{ii} \neq 1$, the derivative of $\Phi(\boldsymbol{x}; \zeta)$ corresponding to $\zeta_i$, denoted by $\nabla_{\zeta_i}\Phi(\boldsymbol{x}; \zeta)$, is a $\Lambda'$-quasi-homogeneous function with $\Lambda' = \frac{1}{1-\Lambda_{ii}}\Lambda$. Second, taking the derivative $\nabla_\alpha\Phi(\boldsymbol{x}; \psi_\alpha(\zeta))$ at $\alpha = 0$, we have:

$$\zeta^T\Lambda\nabla_\zeta\Phi(\boldsymbol{x}; \zeta) = \Phi(\boldsymbol{x}; \zeta). \quad \text{(derivative equation of } \Phi \text{ at } \boldsymbol{x}) \quad (9)$$

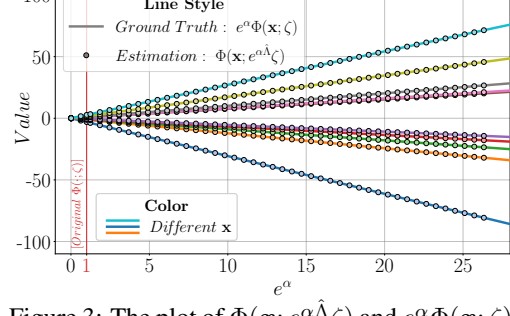

Figure 3: The plot of $\Phi(\boldsymbol{x}; e^{\alpha\hat{\Lambda}}\zeta)$ and $e^\alpha\Phi(\boldsymbol{x}; \zeta)$, with $\hat{\Lambda}$ as the estimation of $\Lambda$. Color distinction signifies diverse input $\boldsymbol{x}$. The coincidence of solid and dotted lines validates the precise estimation of $\Lambda$, because, by definition, an accurate estimation fulfills $\Phi(\boldsymbol{x}; e^{\alpha\hat{\Lambda}}\zeta) = e^\alpha\Phi(\boldsymbol{x}; \zeta)$.

For clear notation, we refer to the above Eq. (9) as the derivative equation of $\Phi$ at $\boldsymbol{x}$. which is a linear equation about $\Lambda$ and the coefficients can be calculated conveniently given the pre-trained model.

As these properties of quasi-homogeneous function are independent of input $\boldsymbol{x}$, we can establish a system of equations by evaluating the derivative equation of $\Phi$ and its higher order derivatives at a set of random samples $\{x_i\}_{i\in[K]}$, e.g.,

$$\begin{cases} \zeta^T\Lambda\nabla_\zeta\Phi(\boldsymbol{x}_1; \zeta) = & \Phi(\boldsymbol{x}_1; \zeta), & \text{(derivative equation of } \Phi \text{ at } \boldsymbol{x}_1) \\ \zeta^T\Lambda\nabla_\zeta\Phi(\boldsymbol{x}_2; \zeta) = & \Phi(\boldsymbol{x}_2; \zeta), & \text{(derivative equation of } \Phi \text{ at } \boldsymbol{x}_2) \\ \quad\vdots & \quad\vdots & \\ \frac{1}{1-\Lambda_{11}}\zeta^T\Lambda\nabla^2_{\zeta\zeta_1}\Phi(\boldsymbol{x}_K; \zeta) = & \nabla_{\zeta_1}\Phi(\boldsymbol{x}_K; \zeta), & \text{(derivative equation of } \nabla_{\zeta_1}\Phi \text{ at } \boldsymbol{x}_K) \\ \quad\vdots & \quad\vdots & \end{cases} \quad (10)$$

Then, the $\Lambda$ can be calculated from it. As a proof of concept, the estimated results of a two-layer fully-connected network with ReLU activation are shown in Fig. 3. The architecture of the network is $\boxed{Linear(2, 10)} \rightarrow \boxed{ReLU()} \rightarrow \boxed{Linear(10, 1)}$.

**Determine $\alpha$.** To compute $L_{duality}$, it is necessary to determine the $\alpha$. Shown in the proof of Theorem 1 [Kunin et al., 2023], the value of $\alpha$ depends on the minimum classification margin $q_{min}$ of the (normalized) neural network on the training dataset, in the case of a multi-class problem:

$$q_{min} = \min_{i\in[N]} \min_{c\in[C]/\{y_j\}} \left[\Phi_{y_i}(x_i; \bar{\zeta}) - \Phi_c(x_i; \bar{\zeta})\right]. \quad (11)$$

Therefore, the value of $\alpha$ depends on the entire training dataset, which is not accessible. To avoid the accumulation of errors in estimating $q_{min}$ using generated samples during training, we directly optimize $\alpha$ as a trainable parameter.

**Summary of the method.** In short, the optimization problem used to train the generator $g(\cdot, \cdot; \theta)$ can be written as:

$$\min_{\theta, \eta, \alpha} L(\theta, \eta, \alpha) = \min_{\theta, \eta, \alpha} L_{stationarity}(\theta, \eta) + \beta L_{duality}(\theta, \alpha). \quad (12)$$

The training procedure is summarized in Alg. 1 in the Appendix.

**An extension to multiple classifiers.** We have previously discussed a method using a single classifier to train a generator. Here, we present an extension of our approach to employ multiple classifiers for training a single generator. Given $T$ classifiers $\{\Phi^{(t)}\}_{t\in[T]}$, we encode the training

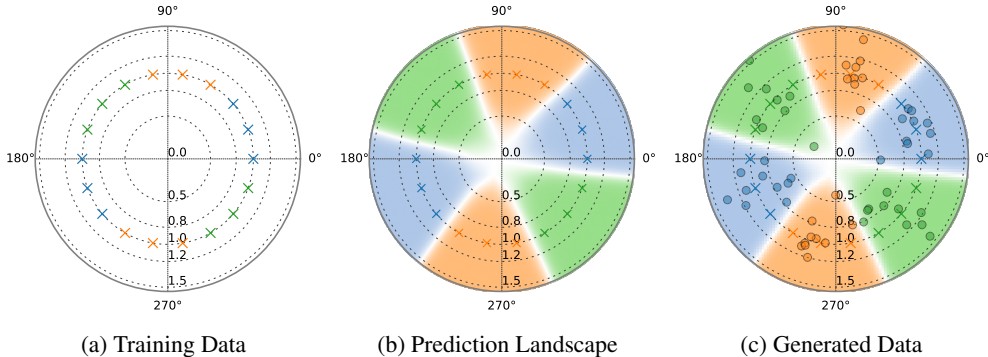

| (a) Training Data | (b) Prediction Landscape | (c) Generated Data |

Figure 4: An illustrative 2D example showcases our proposed method. Figs. 4a to 4c are the training data employed for the classifier, the classifier's learned prediction landscape, and the generated samples using the learned generator, respectively.

data information from distinct classifiers into one generator by incorporating the classifier index $t$ as an input of the generator. By providing random noise $\epsilon$, label $y$, and classifier index $t$ as inputs, $x$ is generated as $x = g(\epsilon, y, t; \theta)$. During the training process, we compute the loss for each classifier, $L^{(t)}(\theta, \eta, \alpha^{(t)})$, and aggregate them as the final optimization objective. For each classifier, we optimized a distinct $\alpha$ and use $\alpha^{(t)}$ to denote the $\alpha$ for the $t$-th classifier. We utilize a unified network $h : \mathcal{R}^d \times \mathcal{Y} \times [T] \to \mathcal{R}^{|\mathcal{Y}|}$ to compute the KKT multipliers, which also includes the classifier index as an input. The optimization objective for the training process can be formulated as:

$$\min_{\theta, \eta, \{\alpha^{(t)}\}_{t \in [T]}} \sum_{t \in [T]} L^{(t)}(\theta, \eta, \alpha^{(t)}). \tag{13}$$

Upon completion of training, generating samples requires an additional step to sample a suitable classifier index. Let $\mathcal{T}_y$ denote the set of classifier indices, where the indices in the set correspond to classifiers whose training set contains samples with label $y$. To generate a sample $x$ with label $y$, we first sample a classifier index $t$ from $\mathcal{T}_y$, and then utilize $t$, along with $y$ and random noise $\epsilon$, as inputs to the generator to produce the sample.

The algorithm for training a generator using multiple classifiers is summarized in Alg. 2 in the Appendix.

## 5 Experiments

### 5.1 An Example of 2D Synthetic Data

In this subsection, we employ a two-dimensional example to analyze the proposed approach. We evenly generate 18 data points on a unit circle, and categorize them into three classes to create our training dataset $D$. The distribution of $D$ is plotted in Fig. 4a, with distinct colors representing different categories. Utilizing this data, we train a three-layer fully connected network $\Phi$ as the classifier. The prediction landscape of $\Phi$ is displayed in Fig. 4b. The color of the region represents the category predicted by the classifier for samples within that region. The white areas represent the classifier's decision boundary. Given $\Phi$, we train a generator $g$ using the proposed method. The samples generated by $g$ are plotted in Fig. 4c. Despite the presence of certain noise, the distribution of the generated data aligns consistently with that of the classifier's training data.

Leveraging this two-dimensional example, we further analyze our proposed method of training a generator using multiple classifiers. We evenly split the previous training dataset into $D_1$ and $D_2$, plotted in Figs. 5a and 5f, respectively. Using these two subsets of data, we train two classifiers, $\Phi_1$ and $\Phi_2$, and subsequently train two generators, $g_1$ and $g_2$. Figs. 5c and 5h display the data generated by $g_1$ and $g_2$, respectively. Here, during the training of $g_1$, we solely use $\Phi_1$, and during the training of $g_2$, we solely use $\Phi_2$. Obviously in Figs. 4b and 5g, owing to the generalization ability of neural networks, $\Phi_1$ and $\Phi_2$ learn additional categorization capabilities beyond the training data, with prediction areas with the same color surpassing the region covered by the training data. Contrarily, the samples generated by training $g_1$ and $g_2$ congregate around the actual data points, effectively

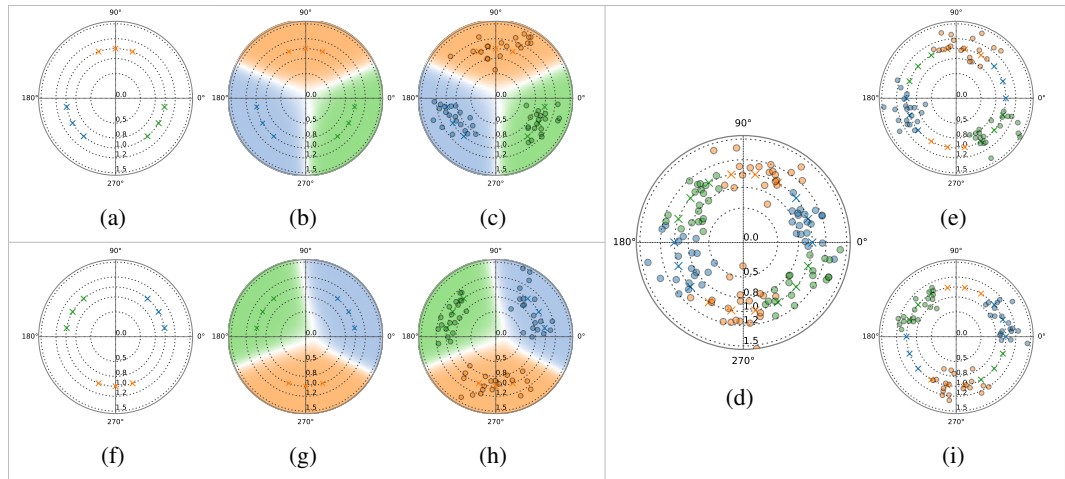

Figure 5: An illustrative 2D example showcases the process of using two pre-trained classifiers to train a generator. Figs. 5a to 5c and Figs. 5f to 5h are two groups of training data, classifier's learned prediction landscape, and generator's generated samples. Fig. 5d shows the generated samples of the generator trained using two classifiers. Figs. 5e and 5i are the generated samples of the generator in Fig. 5d with fixed classifier index.

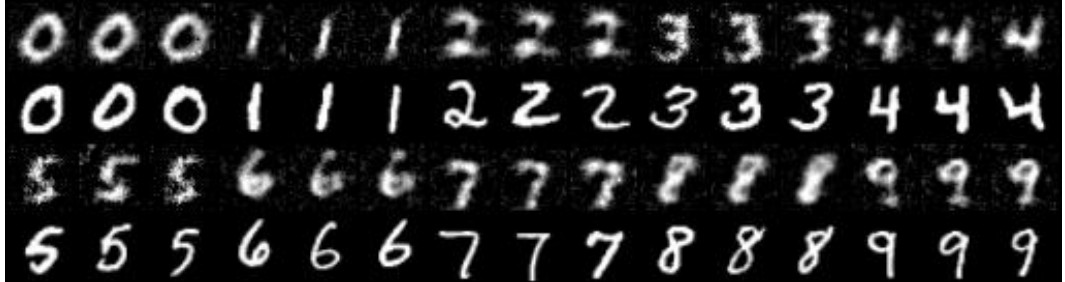

Figure 6: Generator-produced samples. The generator is trained using a single classifier trained on the MNIST dataset.

recovering the original distribution of the training data. Subsequently, by employing the extension of the proposed method, we train a generator $g_{1+2}$ using $\Phi_1$ and $\Phi_2$ together. As shown in Fig. 5d, $g_{1+2}$ possesses the generative capabilities of both $g_1$ and $g_2$, capable of generating samples on the entire training data $D$. We further fix the classifier index of $g_{1+2}$ to either 1 or 2, randomly sample noise and category labels, and observe the data generated by $g_{1+2}$. As depicted in Figs. 5e and 5h, $g_{1+2}$ is also capable of independently generating data belonging to either $D_1$ or $D_2$.

## 5.2 Image Generation

In this subsection, we showcase the experimental results on the MNIST [Lecun et al., 1998] and CelebA [Liu et al., 2015] datasets. More results and implementation details are left in the appendix. For the MNIST dataset, we set up a classification task corresponding to the digits 0-9 with 500 training data (50 images per class) randomly sampled from the original training set. For the CelebA dataset, we utilized various binary attributes to construct binary classification tasks on facial images, for example, distinguishing between males and females. For each task, we randomly sampled 100 images (50 images per class) from the original training dataset and resize them to 32x32 to be our training dataset.

We employed a three-layer fully-connected network with a ReLU activation function and batch normalization as the classifier for the aforementioned classification tasks. The networks were trained until the classification loss converges using full batch gradient descent, which ensures the parameters are close to the convergence point required in the theory of Maximum-Margin Bias.

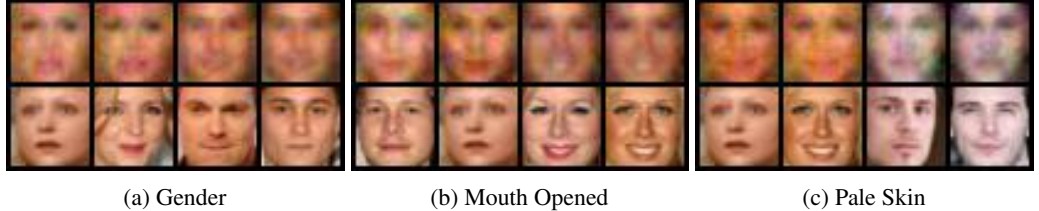

(a) Gender         (b) Mouth Opened         (c) Pale Skin

Figure 7: Generator-produced samples. The generator is trained using a single classifier trained on the CelebA dataset. The captions of the subfigures indicate the attributes used as the label.

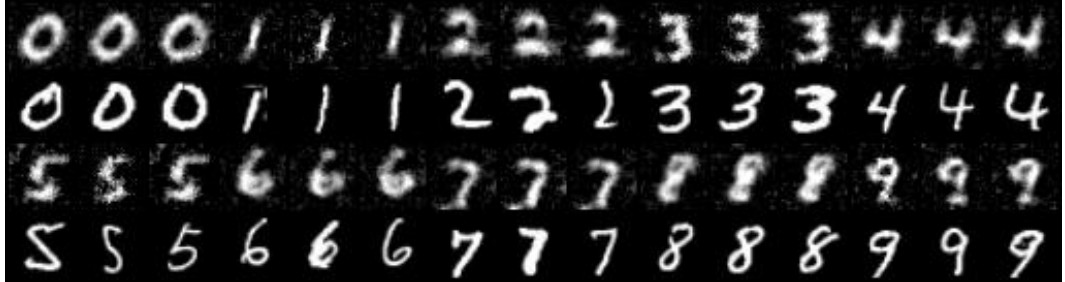

Figure 8: Generator-produced samples. The generator is trained using two classifiers trained on the MNIST dataset.

We use generators composed of three fully-connected layers followed by three transposed convolution layers, with ReLU activation function and batch normalization. Network parameters were initialized using Kaiming initialization [He et al., 2015] and trained for $50,000$ epochs. The batch size and learning rate were set as hyperparameters and optimized via random search. In order to control the noise in generated images, we also utilized total variation loss in pixel space as a regularization term.

The generated results on MNIST and CelebA are shown in Figs. 6 and 7. Odd rows present the generated images, and even rows present the images from the training dataset that are closest to the generated images. The distance between images is measured by the SSIM metric. As shown by the results, the generator trained using our method is capable of generating digits and facial images, even though it has never been exposed to images of digits or faces.

To validate the extended method we proposed for multiple classifiers, we partitioned the aforementioned digit classification dataset into two subsets including digits $0-4$ and digits $5-9$, respectively, and trained two classifiers separately. We then employed our method to train a single generator using both classifiers. Fig. 8 showcases the final images generated. The trained generator successfully integrates information from both classifiers, being capable of generating all digits from 0-9.

## 6 Conclusion

In this research, we investigate a pioneering task: training a generator directly utilizing a pre-trained classifier, devoid of training data. Based on the maximum margin bias theory, we present the relationship between pre-trained neural network parameters and the training data distribution. Consequently, we devise an innovative loss function to enable the generator's training. Essentially, our loss function requires the generator to guarantee the optimality of the parameters of the pre-trained classifier under its generated data distribution. From a broader perspective, the reuse and revision of pre-trained neural networks have been a widely studied direction. [Ma et al., 2023, Fang et al., 2023a,b, Yu et al., 2023, Yang et al., 2022a,b] Our method offers a novel direction for leveraging pre-trained models.

## Acknowledgment

This project is supported by the National Research Foundation, Singapore under its AI Singapore Programme (AISG Award No: AISG2-RP-2021-023), and the Singapore Ministry of Education Academic Research Fund Tier 1 (WBS: A0009440-01-00).

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
