# OpenReview forum: "Generator Born from Classifier"
_NeurIPS.cc/2023/Conference — NeurIPS 2023 poster_

### Official Review · Reviewer_LZcP · 2023-07-05

**Soundness:** 2 fair
**Presentation:** 2 fair
**Contribution:** 3 good
**Rating:** 7
**Confidence:** 2

**Summary:**

This paper tackles the problem of reconstructing an image generator, without relying on any data. The authors propose a learning paradigm in which the generator is trained to ensure that the convergence conditions of the network parameters are satisfied over the generated distribution of the samples.

**Strengths:**

1. The paper addresses an important and challenging problem.
2. The method appears to be novel.

**Weaknesses:**

1. The evaluation seems to be weak. the authors have performed the qualitative evaluation alone. The experimental section seems to be incomplete without quantitative evaluation using metrics such as precision, recall, and FID.
2. Related works on Data free knowledge distillation are missing.

**Questions:**

1. Why does the generator generate realistic-looking samples even though it could cheat by learning to generate adversarial examples?
2. Are there any experimental validations/theoretical justifications to confirm if the model is not just reconstructing the training data examples?
3. Can the generator capture all the modes of the training data distribution sufficiently well?

**Limitations:**

The authors haven't discussed the potential negative social impact of their work, in particular, these methods could be used to recover the training data which may cause privacy concerns.

---

> ### Author Rebuttal · Authors · 2023-08-10
>
> ### **Why does the generator generate realistic-looking samples even though it could cheat by learning to generate adversarial examples?**
> We appreciate the reviewer's question. Indeed, our loss design inherently penalizes the generation of adversarial samples. Since our loss is derived from the KKT conditions of the optimization problem stated in Theorem 1, it mandates the training of a generator such that the samples it produces ensure the optimality of the classifier network parameters. An intuitive explanation is as follows:
>
> Our approach requires the generator to address this question: "How should I generate data to ensure that when a classifier is trained using the data I generate, the resulting classifier parameters precisely match the given classifier's parameters?"
>
> Given that overparameterized neural networks tend to overfit their training dataset, the generated data distribution needs to closely resemble the real data distribution to guarantee the optimality of the classifier network parameters. As a result, our method is capable of producing realistic-looking samples that are close to the true training data distribution.
>
> ### **Are there any experimental validations/theoretical justifications to confirm if the model is not just reconstructing the training data examples?**
> We appreciate the reviewer's question. To demonstrate that we are not merely reconstructing the original data, we conducted an attribute editing experiment and show the results in Fig. 1 in the attached PDF file. By manipulating the input noise, we generated continuously rotating digits, which are not included in the original dataset. This experiment demonstrates that we can indeed generate samples not present in the original dataset, akin to a generator model trained directly on the data.
>
> ### **Can the generator capture all the modes of the training data distribution sufficiently well?**
> We appreciate the reviewer pointing out the issue with our method in terms of capturing all the modes equally. As shown in Fig. 4 in our manuscript, our method does not perform well in reconstructing certain modes (certain categories). We hypothesize that there might be two reasons leading to this phenomenon. First, different categories may have varying complexities in their data distributions, making it more challenging for the generator to produce certain categories. Second, the generator might inherit the classifier's bias. If the classifier does not fit well for certain categories, the information about that category in the classifier might be limited or inaccurate. This, in turn, would affect the generator's performance on that particular category.
>
> ### **Evaluation**
> We appreciate the reviewer's suggestion to include more evaluations. We have calculated the FID and the precision and recall. The results are shown in Tab. 1.
>
> ### **Related Work**
> We appreciate the reviewer's suggestion to include related work on Data-Free Knowledge Distillation. This will be incorporated into our revised version.
>
> ### **Potential Social Impact**
> We appreciate the reviewer pointing out the potential negative social impact of our method in terms of privacy leakage. We mentioned the risk our method poses to privacy protection in lines 319-320. In our revised version, we will further expand the discussion on this potential negative social impact.

---

> > ### Comment · Reviewer_LZcP · 2023-08-14
> >
> > Thanks for the detailed response.
> >
> > [Are there any experimental validations/theoretical justifications to confirm if the model is not just reconstructing the training data examples?]
> >
> > The attribute editing experiment seems to be quite insightful. A quantitative evaluation would also  be quite helpful in this aspect.
> >
> > Regarding performance
> >
> > From the table 1, it seems that quality of the generated images is inferior to that of Li, 2022. I assume the advantage would be then only be then with respect to complexity. Am I correct?
> >
> > Apart from these, the authors have clarified all my concerns. Feedback on these additional points would be appreciated.

---

> > > ### Author Response · Authors · 2023-08-15
> > > **Response to Reviewer LZcP**
> > >
> > > ### **Are there any experimental validations/theoretical justifications to confirm if the model is not just reconstructing the training data examples?**
> > > #### **Quantitative Evaluation**
> > > We appreciate the reviewer's question. To quantitatively assess whether our method can generate images distinct from the training data, we measured the mean-max Structural Similarity Index (mean-max-SSIM). The computation for this metric is as follows: (1) For one generated image, we measure its SSIM against every image in the classifier's training dataset and select the maximum SSIM. This step measures the similarity between the generated image and the most similar image from the training set. (2) For a set of generated images, we perform step (1) for each generated image and then average the resulting max-SSIM values to obtain the mean-max-SSIM. A lower mean-max-SSIM indicates that the generated images, even when compared to their closest counterparts in the training set, are relatively distinct, suggesting the generation of novel data outside the original dataset.
> > >
> > > We compared our method with Haim et al., 2022, which is designed for reconstructing the training dataset. The FID scores of our method and Haim et al., 2022 are comparable, which are shown in Tab. 1 in the attached PDF file. However, our method exhibits a lower mean-max-SSIM, indicating its capability to produce novel images distinct from the original dataset.
> > >
> > > | 　     |  **Ours**  | **Haim et al.,2022** |
> > > |  :-: | :-: | :-: |
> > > | mmSSIM | 0.012±3e-4 |      0.020±1e-4      |
> > >
> > >
> > > ### **Comparison with Li, 2022**
> > > We thank the reviewer for the question. To clarify further, our approach differs from Li, 2022 mainly in two aspects. Besides the complexity aspect the reviewer mentioned, the other distinction is that Li, 2022 necessitates prior information from the original dataset to guide image generation. By contrast, our method operates without this information.
> > >
> > > Specifically, Li, 2022 requires the knowledge of the mean of training images and incorporates the norm between the generated images and the mean of training images as a regularization term in the loss function. We experimentally observe that the method in Li, 2022 heavily relies on this prior knowledge, without which, the method tends to produce adversarial samples.

---

> > > > ### Author Response · Authors · 2023-08-18
> > > > **Response to Reviewer LZcP**
> > > >
> > > > We greatly appreciate the reviewer's questions and kindly mention that we have a prior response addressing the questions. In this follow-up response, we present additional new experimental results as an enrichment of our previous answers
> > > >
> > > > We are also more than willing to engage in a detailed discussion and address any remaining questions the reviewer might have.
> > > >
> > > > ### **Are there any experimental validations/theoretical justifications to confirm if the model is not just reconstructing the training data examples?**
> > > > #### **Quantitative Evaluation Cont.**
> > > > In our previous response, we employed the SSIM metric in the **pixel space** to quantitatively demonstrate that our method can indeed generate images not present in the original dataset. We further validated this in the **feature space**. Specifically, we computed the mean-min $l_2$ distance (mean-min-$l_2$) for the features. The calculation for this metric is as follows: (1) We utilized a backbone neural network to compute the latent features of all images in the classifier's training dataset. (2) For a generated image, we first computed its feature using the backbone network, then measured the distance between this feature and the features of each image in the training set, selecting the minimal value as the min-$l_2$. Consistent with our previous idea, this step measures the distance between the generated image and its most similar counterpart in the training set. (3) For a set of generated images, we perform step (2) for each generated image and then average the resulting min-$l_2$ values to obtain the mean-min-$l_2$. A higher mean-min-$l_2$ also shows that the generated images are different, even compared to the closest examples from the training set. This hints at the generation of data that are not in the original set.
> > > >
> > > > We use Inception-v3 as the backbone network, which is commonly used for computing metrics such as FID and ISC. We compare our method with the method presented in Haim et al., 2022 as in the previous experiment. The experimental results in the table below are consistent with our previous findings, with our method exhibiting a higher mean-min-$l_2$. This further substantiates our claim that our approach can generate images that do not exist in the training set. Moreover, we replicated the experiment using the feature from different neural network layers, and the results consistently supported our findings.
> > > >
> > > > |         　         | Layer Name | Layer Depth |    Ours   | Haim et al.,2022 |
> > > > |:------------------:|:----------:|:-----------:|:---------:|:----------------:|
> > > > |  Feature mm-$l_2$ of |  MaxPool_2 |   Shallow   | 1.12±3e-2 |     0.98±3e-2    |
> > > > | Feature mm-$l_2$ of  |  Mixed_6e  |      ↓      | 0.91±2e-2 |     0.82±9e-3    |
> > > > |  Feature mm-$l_2$ of |   AvgPool  |     Deep    | 13.0±9e-2 |     12.5±9e-2    |

---

> > > > > ### Author Response · Authors · 2023-08-21
> > > > > **Response to Reviewer LZcP**
> > > > >
> > > > > We appreciate Reviewer LZcP for the comprehensive review and insightful feedback. If possible, please let us know whether we have addressed all the concerns. Should the reviewer have other comments, we are more than happy to address them.

---

> > > > > > ### Comment · Reviewer_LZcP · 2023-08-21
> > > > > >
> > > > > > Thank you for the effort put into the rebuttal.  My concerns have been addressed and I would like to increase my score.

---

### Official Review · Reviewer_5Ctm · 2023-07-05

**Soundness:** 4 excellent
**Presentation:** 3 good
**Contribution:** 3 good
**Rating:** 8
**Confidence:** 4

**Summary:**

This paper addresses a novel and intriguing issue, namely, training a generator reliant on a pre-trained classifier, rather than extensive training data. The authors propose an innovative solution, fundamentally aimed at enabling the generator's training process to continuously extract and leverage information about the dataset distribution from the classifier's parameters. Based on the theory of Maximum-Margin Bias of gradient descent, the authors design stationarity loss and duality loss to ensure that the distribution of data generated by the trained generator guarantee the optimality of the pre-trained classifier. As an expansion, the paper also proposes an algorithm to collectively train a generator using information from multiple pre-trained classifiers. The proposed learning paradigm is empirically validated through experiments in various image generation tasks.

**Strengths:**

This paper investigates an intriguing and innovative task: training a generator using information from the parameters of pre-trained classifiers, rather than relying on training data. Given the vast number of pre-trained models our community has accumulated, I view this as a task of substantial value with potential impact.

The paper clearly outlines its motivation and problem definition, and offers a technically sound and concrete method, underpinned by the theory of Maximum-Margin Bias of gradient descent. The paper processes a clear logical structure and reasonable organization. The proposed method and the technical details is validated through several convincing proof-of-concept experiments, as shown in Fig. 3, 4 and 5. The experiments across multiple image generation tasks exhibit promising results.


**Weaknesses:**

Considering that this is a new attempt in this task, the experimental setup is understandably simple. Both the generator and classifier architecture utilized are elementary compared to the practice today. Besides this, I have some minor comments, which I have listed as questions below.

**Questions:**

1.	Ln 226 mentions that the construction of the system of equations depends on certain random samples. Are these random samples drawn from the original training dataset? If the solution for \Lambda relies on samples from the training dataset, does it imply that the method proposed in the paper intrinsically presumes \Lambda to be known?
2.	The dataset used for training the classifier in the paper is relatively small, consisting of only 500 samples. What is the reasoning behind this setup? How would increasing the size of the training dataset impact the results?
3.	Typo
a.	Ln. 241, the set of classifiers is missing {}


**Limitations:**

Owing to the theory of Maximum-Margin Bias, the proposed method has limitations regarding the architecture of the pre-trained classifier and the pre-computation of \Lambda. However, the authors have adequately addressed these challenges.
The paper also points out potential privacy risks that may arise from the application of the proposed method, as well as outlining some potential applications.

---

> ### Author Rebuttal · Authors · 2023-08-10
>
> ### **How the random samples are generated when calculating Lambda?**
> We appreciate the reviewer's question. The definition of a quasi-homogeneous function is independent of the dataset; here, the random samples refer to random noise, not samples selected from the original dataset. For instance, if the classifier's input is an image of size 3xHxW, sampling random noise of size 3xHxW as the classifier's input is sufficient for determining Lambda. Therefore, the value of Lambda does not depend on the training dataset but solely on the pre-trained neural network. Given a neural network, Lambda can be determined using the method described in the 'Determine Lambda' section of the paper, so it does not need to be known in advance.
>
> ### **Concern about the Number of Training Samples**
> We appreciate the reviewer's inquiry about our experimental setup. Since the validity of the Maximum-Margin Bias theory depends on the convergence of the pre-trained network, we used a small dataset to ensure the classifier's convergence. We conducted experiments using larger datasets, and the results are presented in Tab. 5 in the attached PDF file. As the number of training data increases, it offers more information about the data distribution, leading to an enhancement in the quality of generated images.
>
> ### **Typos**
> We appreciate the identification of our typos, and we will correct them.

---

> > ### Comment · Reviewer_5Ctm · 2023-08-17
> > **Thanks**
> >
> > Thanks for your rebuttal. I have no more questions and keep my positive score.

---

### Official Review · Reviewer_KxBC · 2023-07-05

**Soundness:** 2 fair
**Presentation:** 1 poor
**Contribution:** 1 poor
**Rating:** 3
**Confidence:** 4

**Summary:**

The paper propose a method to generate image samples from a trained neural classifier. It proposes a loss to train an image generator where the loss is based on recent results from the realm of the implicit bias of gradient descent of quasi-homogenous functions. It shows empirical results on 2D data and on models trained on MNIST and CELEBA datasets.

**Strengths:**

- Extension of reconstruction to quasi-homogenous neural networks

- Trying to create a generator by using a fixed prior on top of the reconstruction scheme

**Weaknesses:**

I think that the main weakness is the novelty in the paper. The proposed paper seems like an extension of Haim et al. 2022 to multiclass quasi-homogenous network, using a "generative" prior. However:
- Note that a multiclass extension to Haim et al was done in Buzaglo et al. 2023 ICLR workshops (https://openreview.net/forum?id=SBstNm4OajH), which is somewhat concurrent).
- the extension of the reconstructoin loss to quasi-homogenous networks is very slim - technically speaking, the only difference is the introduction of $\Lambda$ in the loss. If this is truly necessary it should be supported by some comparisons to Haim et al. which also showed results on non-homogenous networks. Such comparisons are not provided in the paper.
- The generative prior is interesting, but there is no evaluation in the paper for a generative model. All in all, the results are quite similar to those of Haim et al. only it seems that they were not fully converged (what is the difference between a "bad" reconstruction and a "generated sample"?)

The paper is poorly written - terminology is cumbersome and hard to follow, which makes it difficult to understand key components of the paper. Many important technical details are missing from the paper.

Some comments in introduction and related section are unclear or irrelevant.



**Questions:**

Evaluation is lacking:
- the main claim of the paper is the proposed generator that is being born from a classifier. But there are no evaluation whatsoever of generative models. Many common metrics for evaluating generative models exists like FID or IS (for GM that models a classified datasets), but non of these are provided.
- There are only very few visual examples in the paper, and no further analysis
- The method proposes creating a generative model -  what is the point in comparing the outputs of the model to their nearest neighbors (Fig. 6,7,8)? (there are established methods for evaluating generative models, see above)
- No comparison to baselines: basically, the results look like a lesser version of the Haim et al. results. What happens if the paper uses Haim et al. reconstruction loss on their trained models? The only concern may be of the multiclass models - such an extension is shown in Buzaglo et al. 2023, which is somewhat concurrent to this submission. However, the current submission can show a comparison of their proposed method to that of Haim et al. only on binary classifiers.
- Some other baselines could be other works that try to "turn" a classifier to generator:
1) Use Classifier as Generator, Li 2022
 2) Traditional Classification Neural Networks are Good Generators: They are Competitive with DDPMs and GANs, Wang and Torr, 2022


Paper is poorly written. Main parts of the paper are difficult to follow and understand:
- The main contribution in the paper is the training of a generator (as assumed from the title). However, the discussion of the generator itself is somewhat hidden in lines 185-186. It took me a while to figure out that this is the so called generator that the paper discusses. I suggest emphasising this element much more.
- the discussion of determininig $\Lambda$ (in lines 209-229) - I can't say that I understood the derivation. why is $\Lambda' = \frac{1}{1-\Lambda_{ii}}\Lambda$? I could not understand the two properties. I also did not understand the plot showni in Fig. 3
- discussion in lines 201-205 - why is this network h and ReLU is necessary? this part is very cryptic and not clear (not the mechanism itself, nor the motivation or intuition behind this design choice).
- What is the difference between Eq. (7) and Eq. (12) - it seems like the exact same equation
- Why is the extension to multiple classifiers (lines 239seqq.) necessary? what is the motivation for that? I also could not understand the mechanism in this paragraph.


Many important technical details are missing from the paper:
- what is the architectures of the trained models? said to be quasi-homogenous
- what is the architecture of the generator g?
- what is the architecture of the network h? (in line 202)


Some comments in introduction and related section are unclear or irrelevant:
- "training process does not rely on any training data" (line 44 and 56) is a bit misleading - it relies on the data on which the classifier was trained. (also it seems from eq.4 that the number of training samples N is assumed to be known).
- "GAN require an additional classifier" (line 71) - what is the meaning here? unconditional GANs do not require additional classifiers
- "the classifier is trained concurrently with the generator" (line 73) - is this referring to the discriminator? because this is not the classifiers in the sense that are used in the paper. The current paper submission discuss "usual" classifiers (that solve the classification problem of a supervised classification dataset) and discriminators are classifying between true and fake samples.
- References in line 85 - "model inversion... given the output and the trained model" - Yin et al. 2020 do not assume the outputs but rather infer training samples from the trained batchnorm statistics. Gal et al. 2022 is really irrelevant here (textual "inversion" tries to find a token in the input of the text encoder that corresponds to a given set of images. It has nothing to do with training samples reconstruction).


Minor:
- line 207 and supplementary 68 - L_{lagrange} should be L_{stationarity}?

---

> ### Author Rebuttal · Authors · 2023-08-10
>
> ### **Comparison with [1] Haim et al. 2022 and [2] Buzaglo et al. 2023**
>
> We appreciate the reviewer's comparison of our work with [1] and [2]. We kindly direct the reviewer to the global response, where we systematically compare ours with [1] and [2].
>
> #### **The importance of Lambda**
> Lambda's introduction is vital for complex classifiers. While [1] mandates homogeneous networks, we allow quasi-homogeneous ones. This necessitates solving for Lambda in each network and integrating it into the loss of generator training.
>
> ### **Evaluation**
> #### **Comparisons with [3] Li, 2022 and [4] Wang and Torr, 2022**
> We appreciate the reviewer's suggestion to compare our method with those in [3] and [4]. We kindly direct the reviewer to the global response, where we systematically compare ours with [3] and [4].
>
> #### **Quantitative Metrics**
> Thanks. We compared the quantitative metrics of our method with [1] and [3] and reported in Tab. 1 in the submitted PDF for the results.
>
> Due to the fact that [4] did not release their source code (the provided GitHub link directs to a non-existent repository), we attempted to implement their work. However, given the tight timeframe for the rebuttal, we were unable to fully reproduce the results reported in their paper. As a result, we have not included the results from [4] at this time.
>
> #### **Further Analysis**
> We are grateful for the reviewer's suggestion about further analysis. We conducted an attribute editing experiment and show the results in Fig. 1 in the attached PDF file. By manipulating the input noise, we generated continuously rotating digits, which are not included in the original dataset, akin to a generator model trained directly on the data.
>
> #### **The reason for comparing with the nearest neighbors**
> We appreciate the question posed by the reviewer. Our comparison with nearest neighbors serves merely as a reference to illustrate the perceptual quality of the images generated by our method. By comparing with real data, it's evident that our generator produces images with good quality.
>
> ### **Writing**
> We greatly appreciate the reviewer's suggestions and will make modifications accordingly.
>
> 1. We will highlight the sections about the generator.
>
> 2. We will further clarify our discussion on solving for Lambda.
> * Solving for Lambda involves two steps: (1) constructing a system of linear equations about Lambda and (2) solving the system.
> * Two properties used to construct the system are introduced in Section 3 of "The Asymmetric Maximum Margin Bias Of Quasi-Homogeneous Neural Networks".
> * We will include the derivative of Lambda' in the revised version.
> * The purpose of Fig. 3 in the manuscript is to verify whether the Lambda obtained by our method accurately estimates the true Lambda of the network. According to the definition, given a quasi-homogeneous model $\Phi$, we have $\Phi(x;e^{\alpha \Lambda}\zeta) = e^{\alpha}\Phi(x;\zeta)$. If the estimated Lambda is correct, then substituting it into the left-hand side of the equation, the equation should still hold. In Fig 3, we plotted the values of the right-hand (solid line) and the left side of the equation (dotted line) with estimated Lambda for different $x$ and $a$. The overlap of the two lines indicates that our estimate is accurate.
>
> 3. The introduction of $h$ and ReLU in lines 201-205 is for the estimation of the KKT multiplier $\mu$. Given a pair of generated data $x$ and label $y$, $\mu'=h(x,y;\eta)$.
> Since the KKT multiplier must be non-negative, we let $\mu=ReLU(\mu')$.
>
> 4. Eq. 7 and 12 represent the loss function used for training and the optimization problem used for training, respectively. In the revised version, we will consolidate Equations 7 and 12 into a single equation.
>
> 5. The extension to multiple classifiers.
> * Integrating multiple classifiers for training a generator can lead to a more powerful generator. Firstly, different classifiers encounter different data (see the case in Fig 5 in our manuscript), and combining knowledge from multiple classifiers can enhance the diversity of generated samples. On the other hand, using multiple classifiers is also a means to improve the performance of the resulting generator. As shown in Tab 4 in the attached PDF, a generator obtained using two smaller classifiers performs better than one obtained using a lrger classifier. This is because the two classifiers each fit the data they encounter without interfering with each other, thereby preserving the information from the training dataset more effectively.
> * We provide an example of our mechanism: The input to the generator is (noise, label, classifier index). If there are two classifiers, trained on even and odd numbers from MNIST, respectively. To generate the first even digit '0', the input to the generator is (noise, 0, 0); to generate the second odd digit '3', the input is (noise, 1, 1).
>
> ### **Technical Details**
> We thank the reviewer. We indeed include the structure of the classifier and that of the generator in ln 291-294 and ln 295-299, respectively. Specific architectures of the classifier, generator, and net $h$ are shown in Fig. 2 in the submitted PDF. We will also release our code.
>
> ### **Other Comments in Introduction and related work**
> We appreciate the reviewer's detailed suggestions, and we will make modifications and add explanations accordingly.
>
> 1. What we intend to convey here is that the training process of the generator does not require the use of data, but merely a pre-trained classifier and the number of training samples, which does not contain information about data distribution.
>
> 2. In lines 71 and 73, the classifier we mention is the discriminator, as the discriminator is indeed a binary classifier.
>
> 3. We will revise rigorously the discussion about model inversion in the related work based on your comments.
>
> ### **Typos**
> We thank the reviewer for pointing out our typos, and we will correct them.

---

> > ### Comment · Reviewer_KxBC · 2023-08-16
> >
> > I sincerely thank the authors for their elaborated rebuttal.
> >
> > As it seems by looking at the FID/ISC scores, the results are not very different from Haim et al. 2022, and both are pretty "bad" for a generator (very high FID). The way I see it, the main reasons are:
> > - I'm not convinced that the proposed method in the current submission really result in different outputs than simply using Haim et al. 2022 (without the estimation of the lambda etc.).
> > - If the "definition" of a generative model is to model a certain distribution, then using the implicit bias results of Lyu&LI 2019, Kunin et al. 2023 just doesn't make sense. Why would inverting the classifier result in a generative model in the first place? Sample reconstruction is different than modeling a distribution. Simply showing that some reconstruction methods actually work, is not equivalent to saying that they constitute a generative model. This is evident from the FID results, and is the reason why they are so "bad" for a generative model. I assume that many outputs from the proposed "generator" in the work are outputs that doesn't look at all like images from the dataset, because these outputs are in a way solution to the KKT conditions (of the quasi-homogenous case) - which has nothing to do "in general" with modeling the true distribution of the training set. These "bad" outputs are probably also the reason for the very high FID score.
> >
> > On the good side, I wasn't aware to the difference in the number of parameters that are being optimized. This is pretty interesting, and should be emphasized. I also think that the results on attribute editing are impressive - but the details on how they were produced could be clearer.
> >
> > On the bottom line, I find it hard recommending acceptance for the paper in its current form.
> >
> > It would be much easier if the paper was more clearly presenting the novelty w.r.t to previous works and more carefully comparing to them. If the paper was more clearly written, especially the technical parts, and implementing the rest of the remarks in the reviews (which some of the were answered in the rebuttal). Thanks.

---

> > > ### Author Response · Authors · 2023-08-18
> > > **Response to Reviewer KxBC**
> > >
> > > ### **Further Comparison with [1] Haim et al., 2022**
> > > #### **To address:** ***"by looking at the FID/ISC scores, the results are not very different from Haim et al. 2022 .."*** **and** ***"I'm not convinced that the proposed method in the current submission really results in different outputs than simply using Haim et al. 2022 .."***
> > > We appreciate the reviewer's concern. Our global response and Tab. 1 in the PDF compare our method to [1] regarding perspectives, such as complexity and assumptions. Here, we also compare our outputs with [1]'s.
> > > 1. The objective of [1] is to recover images, and the recovered images are their output. However, our goal is to train a generator. The output of our method should be the trained generator, not the images produced by the generator. This generator can produce perceptually satisfying images not present in the original training data, supporting conditional sampling and attribution editing. These capabilities are indeed beyond the reach of [1].
> > > 2. While our method and [1] might not show significant differences in terms of FID/ISC, our parameter space is smaller, and the image generation process is less complex. This indicates that we achieved comparable FID/ISC with a reduced cost, highlighting an advantage of our method over [1].
> > > 3. The fact that our method and [1]'s approach exhibit no significant difference in FID/ISC,  does not imply that the generated images of our method and [1] are largely similar. As shown in Fig.4 and 7 of the attached PDF, it's evident that our method produces images with clearer outlines and more complete shapes.
> > >
> > > Admittedly, given the ambitious nature of our proposed task and being a pioneering effort in this direction, the qualitative result of our approach is not that impressive. This is to be expected, given the inherent challenges of training a generative model, especially as our training was conducted without accessing any data.
> > >
> > > #### **To address** ***"Why would inverting the classifier result in a generative model ...?"***
> > > We thank the reviewer's question and address it from the following three perspectives.
> > > 1. Feasibility: A generative model is essentially an approximation of the conditional probability $P(x|\epsilon, y)$, and its training relies on knowledge about the data distribution $P_x$. When data is available, information regarding $P_x$ can be acquired by accessing the training data directly. However, in our task, training data is absent and only a pre-trained classifier is available. To tackle this task, we adopt the Maximum-Margin Bias (MMB) theory, which the reviewer kindly referred to as the implicit bias theory, to extract information related to training data embedded within the classifier parameters and then train the generator. The crux of this approach's feasibility lies in the sufficiency of the information extracted by the MMB theory. This sufficiency is supported by the findings in [1], as it's conceivable to train a generator using data reconstructed according to [1].
> > > Such an idea demonstrates the sufficiency of the information the MMB theory can extract, affirming the feasibility of our approach.
> > > 2. Parameterization: The distinct parameterization ensures that our approach trains a generator, rather than merely reconstructing data as in [1]. To reconstruct training images, the learnable parameters in [1] are the image pixels themselves. By contrast, we designed a neural network where the network inputs are random noise and label index, and the network output is the image. The weights and biases of this neural network are our learning targets. While both our work and [1] design the loss function based on the MMB theory, this difference in parameterization directly leads to a functional divergence: [1] yields images, while our method produces a generative model.
> > > 3. Experimental Evidence. As demonstrated by our experimental results, we indeed obtained a generator with the capabilities of conditional sampling and attribution editing, though it currently underperforms in metrics like FID.
> > >
> > > We are also grateful for the question regarding the KKT conditions.
> > > As elaborated in both our manuscript and [1], it's acknowledged that the KKT conditions primarily ensure the extraction of information from data points on the classification margin, which represents only a subset of the training dataset. Consequently, during training, our generator predominantly receives supervision signals from these data points on the margin. For the generation of points outside this margin, the generator depends on its learned generalization ability, which might be a reason for the varied quality of certain generated samples. Nonetheless, it's important to recognize that our model indeed operates as a generator. It transforms from the noise distribution to the real data distribution (or more precisely, to the distribution over a subset of training samples), and it also possesses various functionalities typical of generators, as highlighted above.

---

> > > > ### Author Response · Authors · 2023-08-18
> > > > **Response to Reviewer KxBC**
> > > >
> > > > ### **Attribution Editing**
> > > > We thank the reviewer for the suggestions. The experiment of Attribution Editing mainly demonstrates that our method can indeed generate data not present in the original dataset, and it underscores that the generator we trained possesses a semantically structured latent noise space. The specific procedure is delineated in the following four steps:
> > > >
> > > > 1. Initially, we sample an image from the training dataset and produce two new images $x_{0}$ and $x_{N}$ not in the original dataset by rotating the sample image clockwise and counterclockwise, respectively.
> > > > 2. Through inversion, we find the corresponding noise vectors $\epsilon_{0}$ and $\epsilon_{N}$ for $x_{0}$ and $x_{N}$, respectively.
> > > > 3. We perform linear interpolation between $\epsilon_{0}$ and $\epsilon_{N}$, yielding a sequence of noise vectors $\{\epsilon_{0}, \epsilon_{1}, \cdots, \epsilon_{N}\}$.
> > > > 4. Finally, we use $\{\epsilon_{0}, \epsilon_{1}, \cdots, \epsilon_{N}\}$ and the class label as the input of the generator, to generate a series of images.
> > > >
> > > > ### **Comparison with Other Works**
> > > > We thank the reviewer for the comments. The comparison of our work with the works mentioned by the reviewers is presented in the global response and Tab. 1 of the attached PDF. We are more than willing to make further comparisons if the reviewer suggests other works or if there's a specific perspective from which the reviewer would like us to compare our work with others.
> > > >
> > > > ### **Writing**
> > > > We thank the reviewer for the comments. In the revised version, we will refine and enhance our writing to make the paper clearer and more comprehensible. Particularly for the technical part, we will provide more detailed explanations and clarifications.
> > > >
> > > > ### **Implementing the Rest of the Remarks in the Review**
> > > > We thank the reviewer for the valuable feedback during the review and discussion phases. In the revised version, we will incorporate all these suggestions and enhance our manuscript based on the reviewer's comments.
> > > >
> > > > ### **More Remarks**
> > > > We are more than willing to engage with the reviewer regarding any remaining concerns. We would appreciate it if the reviewer specify the concerns they feel are still unresolved, and we will promptly address them.

---

### Official Review · Reviewer_gUqt · 2023-07-06

**Soundness:** 3 good
**Presentation:** 1 poor
**Contribution:** 3 good
**Rating:** 6
**Confidence:** 3

**Summary:**

The work extends the dataset reconstruction (DR) method [1] of reconstructing the dataset from a classifier to the generative scenario. It provides several extensions to it:
- Instead of reconstructing particular data points, it aims to build a generator for the original dataset.
- Extends to the multi-class setting (from a binary classifier).
- Moves from homogeneous to $\Lambda$-quasi-homogenous assumptions for the NN classifier.

The method is tested on MNIST and CelebA and is able to generate samples which resemble the real ones.

[1] Haim et al. "Reconstructing Training Data from Trained Neural Networks"

**Strengths:**

- The overall idea of obtaining a generator from a classifier is very interesting and potentially very influential.
- The work is genuine and open about its strengths and limitations. It feels like one of the rare works which is "what you see is what you get".
- The method is quite novel and shows some promise to be working. I guess, the current ideas might be extendable to GAN training (or at least Auxilliary Classifier GAN training).
- The source code was provided, which should improve the reproducibility.
- Table 1 with the notations in the appendix was quite helpful.

**Weaknesses:**

- The paper is not self-sufficient and not possible to understand without reading the DR paper [1] first (at least, I failed to). It lacks intuitive exposition for the general audience who do not have vast background on the related work. Also, there are some confusing formulations on their own — e.g., Eq 5 and 6 define the loss term over $\theta$ and $\eta$, but they do not explicitly appear in the equation itself (only implicitly through $x_i$). And at the same, $x_i$ is not uniquely defined — it denotes both real and fake data. Also, sometimes it appears in bold (L123), and sometimes in normal font (e.g., Eq 1b). Also, Equation 12 is more or less a tautological repetition of Equation 7.
- The experimental results do not seem to be strong.  E.g., checking the images in Figure 1b, they are hardly recognizable. While for [1], some restored images look quite reasonable, even considering it's CIFAR10 (a more difficult dataset than CelebA).
- There should be a comparison with standard approaches of generating samples from a classifier (e.g., DeepDream or DeepDream from an adversarially robust classifier [3]). After checking some random DeepDream implementation on github [2] (probably untuned), it's hard to tell which samples are really better.
- Limitations are not discussed

Typos:
- L181: "objective, We" => "objective, we"
- L234: "during the training" => "during training"

[2] https://github.com/ianscottknight/deep-dream-implementation-on-mnist-using-pytorch-hooks/blob/main/1.0-deep-dream.ipynb

[3] https://arxiv.org/abs/1906.09453

**Questions:**

My two biggest concerns are the writing quality and the scalability of the method, but I'm unsure how they could be addressed during the short rebuttal period. Another interesting direction would be exploring adversarially robust classifiers [3] (I guess, adversarially robust training should prevent the landscape from covering spurious minima and hence help the generation quality).


**Limitations:**

Limitations are not discussed at all.

---

> ### Author Rebuttal · Authors · 2023-08-05
>
> ### **Scalability**
> We thank the reviewer for pointing out the concern about the scalability of our method. To address the concern, we scale up the classifier and conduct the experiments. We consider two types of architectures: (1) a 10-layer fully-connected network and (2) a network with 3 convolutional layers and 3 fully-connected layers. The quantitative scores corresponding to different classifiers are shown in Tab. 2 in the attached PDF file. After scaling up the classifier the ISC increases. This indicates that the scale-up of the classifier helps to preserve more information about the training data and thus improves the performance of the generators.
>
> ### **Writting**
> We appreciate the reviewer's comment regarding the writing. We will enhance the clarity in the introduction and method sections by incorporating more intuitive explanations of our approach. The main reason for our current writing style is our desire to gradually derive our method from a theoretical standpoint, which necessitated the introduction of numerous mathematical symbols and concepts. In the revised version, we will provide more intuitive explanations for the mathematical symbols and analyses employed.
>
> ### **Exploration of the Adversarially Robust Classifiers**
> We are grateful for the reviewer's suggestion to explore the direction of adversarially robust classifiers. We have replaced the standard classifiers used in our method with adversarially robust classifiers and conducted experiments accordingly. Tab. 3 compares our method with an Empirical Risk Minimization (ERM) classifier, our method with adversarially robust classifiers, and the method mentioned in [3]. It is evident that our method, when combined with adversarially robust classifiers, indeed enhances the quality of the generated images. However, [3] outperforms our method with an ERM classifier. The reasons for this result are two-fold. First, [3] processes a different setting from ours. It optimizes each image individually, thus having a significantly larger optimization space than our method. We are aiming at learning the parameters of a generator and use it to generate all images. Second, [3] utilizes strong prior information, such as the mean and variance of the samples in each class. However, such prior is not used in our method.
>
> ### **Self-Sufficiency and Prerequisite Knowledge**
> We extend our sincere gratitude to the reviewer for advising us to enhance the self-sufficiency of our paper and to incorporate prerequisite knowledge. We will add a section in the supplementary material providing an intuitive introduction to the background and preliminary knowledge, thereby facilitating a more accessible understanding of our work for our readers.
>
> ### **Further Explanation of Some Formulations**
> We appreciate the reviewer's attention to detail in our manuscript.
> 1. In Eq. 5 and 6, the generated samples $x_i=g(\epsilon, y_i;\theta)$ and the Lagrange multipliers $\mu_ic=ReLU(h(x_i,y_i;\eta)[c])$ are produced by neural network $g$ and $h$, respectively. $\theta$ and $\eta$ represent the parameters of these networks. To avoid overcomplicating the equations, we omitted the specific expressions for $x_i$ and $\mu_ic$ in Eq. 5 and 6. However, we will provide their complete expressions in the revised version.
>
> 2. To distinguish between real data samples and generated samples, we will employ different symbols.
>
> 3. The symbol $x_i$ should always be in bold, and we will ensure this is consistently applied in the revised version.
>
> 4. Eq. 7 and 12 represent the loss function used for training and the optimization problem used for training, respectively. In the revised version, we will integrate Eq. 7 and 12 into a single equation.
>
> ### **Performance and Comparison with [1] Haim et al. 2022**
> We greatly appreciate the reviewer's comparison between our work and [1] Haim et al. We kindly direct the reviewer to the global response, where we systematically compare our approach with that of [1] Haim et al.
>
> Admittedly, given the highly ambitious nature of the proposed task and being the first attempt along this line, the performance of our approach, as expected, is indeed not surprising, since our training does not rely on any data and training a generator is inherently challenging.
>
> ### **Comparison with DeepDream**
> We thank the reviewer for comparing our work with DeepDream. We kindly direct the reviewer to the global response, where we systematically compare our approach with that of DeepDream.
>
> ### **Limitations**
> We appreciate the reviewer's suggestion about the limitation section, which we will incorporate into the revised version. Our method has two primary limitations.
> (1) Estimating Lambda introduces additional computational overhead. Given that our approach requires the classifier to be a quasi-homogeneous model, we need to determine the classifier's Lambda before training the generator. The method we provide in our paper for calculating Lambda can be computationally intensive, especially when the classifier has a large number of parameters.
> (2) Our method exhibits class bias, with significant variations in the generation quality for different categories. We hypothesize that there might be two reasons leading to this phenomenon. First, different categories may have varying complexities in their data distributions, making it more challenging for the generator to produce certain categories. Second, the generator might inherit the classifier's bias. If the classifier does not fit well for certain categories, the information about that category in the classifier might be limited or inaccurate. This, in turn, would affect the generator's performance on that particular category.
>
> ### **Typos**
> We are grateful to the reviewer for pointing out the typos in our manuscript. We will rectify all such errors in the revised version.

---

> > ### Comment · Reviewer_gUqt · 2023-08-14
> > **Response to the rebuttal**
> >
> > I am thankful to the reviewers for providing the rebuttal, clarifications and additional results. I decided to increase my rating to "Weak accept", because I find this submission to be quite novel and non-mainstream, and believe that NeurIPS should be welcoming to the works with unusual ideas even they are not tuned well.
> >
> > My two remaining concerns are:
> > - The quality is not good enough and for some people might look like negative results. You can consider applying the technique on top of existing image generators pre-trained for a different objective to show nicer images (note: please, do not treat it a request for additional experiments, I'm just thinking out load).
> > - In its current form, the paper will be a tough read for the image generation community. Since the results are not striking, people will not invest effort into trying to understand such a difficult work. I increased the rating hoping that this will be improved in the revised version.

---

> > > ### Author Response · Authors · 2023-08-14
> > > **Response to Reviewer gUqt**
> > >
> > > We truly appreciate the reviewer for the recognition of our paper and for the valuable feedback. We will incorporate the suggestions and comments the reviewer provided in the “official review” and “response to the rebuttal” to enhance our manuscript, improve its readability, offer a more intuitive background introduction, elucidate our methodology further, and enrich our experimental results.

---

### Author Rebuttal · Authors · 2023-08-10

Esteemed Senior AC, AC, and Reviewers,

We deeply appreciate the reviewers' and ACs' dedication to reviewing and managing our submission. It is with great pleasure that the reviewers highlighted several strengths of our work, such as the importance and the potential impact of the idea/task/problem [gUqt, 5Ctm, and LZcP], the novelty of our method [gUqt and LZcP], the soundness of our method [5Ctm], the promise of our method to be working [gUqt] and the promising results [5Ctm].

**We have additionally uploaded a PDF file containing all supplementary tables and images for perusal.**

Thanks once again for the valuable comments and consideration.

Respectfully,

Authors of 'Generator Born from Classifier'

---

### **Comparison with Other Methods**
Using the global response, we systematically draw comparisons between our approach and the methods mentioned by the reviewers.

We also provide Tab. 1 in the submitted PDF file as a summary of the comparison, which provides an overview of the differences and similarities among the methods and also includes the quantitative evaluation on the MNIST.

#### **Comparison with DeepDream**
We thank the reviewer gUqt for mentioning DeepDream. DeepDream aims to visualize patterns learned by neural networks, while our method aims to train a generator for the conditional sampling of images. Technically, to generate each image, DeepDream requires a gradient optimization process to maximize a certain activation or network output. By contrast, once our generator is trained, images can be produced by sampling random noise. Thus, (1) DeepDream has larger computational complexity and takes longer to generate each image compared to our method; (2) DeepDream has a larger parameter space.

#### **Comparison with [1] Haim et al. 2022 and [2] Buzaglo et al. 2023**
We thank the reviewer gUqt for comparing our work with [1], and reviewer KxBC for comparing our work with [1] and [2].
By the NeurIPS 2023 policy, our work and [2] are concurrent works.
Herein, we group [1] and [2] together for a comparative analysis with our method.

1. **Task.** Our goal is to train a generator network, which is capable of transforming random noise into data of good perceptual quality.
By contrast, [1] and [2] aim to recover training data. Our task is more challenging in two respects. (1) we aim to generate samples that are not present in the original dataset but have good perceptual quality. (2) our generation process is controllable and supports conditional sampling.

2. **#Categories.**  [1] only supports binary classifiers, while our work and [2] support multi-class classifiers.

3. **Supported Networks.** [1] and [2] only support homogeneous networks. By contrast, our method supports quasi-homogeneous networks, which encompass a broader range of networks. The experiments about non-homogeneous models in [1] only involve fully connected networks with bias terms. The networks in our experiments are more complex and diverse.

4. **Technical Aspect.**: [1] and [2] optimize directly on the pixel space. Specifically, #optimizable parameters equals $BatchSize×ImageSize$. In [1], batch size is set to twice the size of the training set.
For MNIST with 500 images, \#parameters is 1M. However, our optimizable parameters only include the parameters of the generator. In our implementation, it is only 0.18M, which is significantly lower than those in [1] and [2].

Therefore, we conclude that we are addressing a more challenging task than [1] and [2] do, and we are doing so with a more constrained budget in terms of #optimizable parameters.

#### **Comparisons with [3] Li, 2022 and [4] Wang and Torr, 2022**
We thank the reviewer KxBC for mentioning [3] and [4].
[3] is to generate an image $x$ given a pre-trained classifier $\Phi$ and a category $y$ according to
$$
    x = \arg\min_{\hat{x}} L(\Phi(\hat{x}),y) + \beta||\hat{x} - \bar{x}||,
$$
where $L$ is the cross-entropy loss, and the regularization term is the norm between the generated sample $\hat{x}$ and training data mean $\bar{x}$,.

[4] is to generate N images $x^1, \cdots, x^N$ that belong to category $y$ given a pre-trained classifier $\Phi$ according to
$$
    x^1, \cdots, x^N = \arg\min_{\hat{x}^1, \cdots, \hat{x}^N} \sum_{i=1}^N L(\Phi(\hat{x}^i),y) + \beta_1 L_{div}(\hat{x}^1, \cdots, \hat{x}^N) + \beta_2 L_{dist}(\hat{x}^1, \cdots, \hat{x}^N),
$$
where $L$ is the cross-entropy loss, $L_{div}$ is a similarity penalty, and $L_{dist}$ constrains the generated samples and the training data to have the same mean and variance in the feature space.

1. **Technical Aspect.** To generate images, [3] and [4] require multiple gradient descent steps to directly optimize each pixel of every image. Each gradient optimization step entails one forward pass and one gradient propagation through the classifier. By contrast, once our generator is trained, images can be generated by only a single forward pass through the generator. This results in two differences: (1) for the generation of each image, [3] and [4] possess a higher computational complexity and take longer to process; (2) [3] and [4] have more optimizable parameters and more expansive parameter spaces.

2. **Prior Information.** [3] and [4] use prior information about the data for generation. Specifically, the mean used in [3] and the mean and variance used in [4]. However, our method solely relies on a pre-trained classifier and does not use the prior information of the data.

Therefore, we conclude that these methods have a different setting. We are dealing with a more challenging task, and our solution uses a lower complexity.

[1] Haim et al, 2022. "Reconstructing Training Data from Trained Neural Networks".

[2] Buzaglo et al, 2023. "Reconstructing Training Data from Multiclass Neural Networks".

[3] Li, 2022. "Use Classifier as Generator".

[4] Wang and Torr, 2022. "Traditional Classification Neural Networks are Good Generators: They are Competitive with DDPMs and GANs".

---

### Decision · Program_Chairs · 2023-09-21

**Decision:**

Accept (poster)

**Comment:**

This paper introduces a method that trains an image generator from a pre-trained classifier, without using samples in the training set. The proposed training objective is based on the Maximum-Margin Bias theory of gradient descent. The paper received four reviews, with three positive reviews acknowledging the novelty of the problem formulation and the proposed solution. One reviewer (KxBC) raises several concerns against the paper including the limited extension of previous work (Haim et al. 2022) to the multi-class setting, limited evaluation of the generative model (the original draft didn’t compare with baselines nor report standard metrics such as FID), poor writing and presentation of the technical part, and missing details of the model architecture. The authors address most of the concerns during rebuttal, which includes additional comparison with the suggested baselines, additional results of the suggested experiments (e.g., performance when a robust classifier is used), explanation and clarification of the writing, and details of the model architecture. Although reviewer KxBC still expresses reservations regarding the novelty of the paper, given other consistent positive scores and the pioneering efforts in this direction (argued by reviewer gUqt), we recommend acceptance for the paper. However, if accepted, the authors need to address the negative reviews in the revision, add suggested evaluations, and improve the clarity of the writing.